# From hidden hearing loss to supranormal auditory processing by neurotrophin 3-mediated modulation of inner hair cell synapse density

Lingchao Ji[1¤], Beatriz C. Borges[1], David T. Martel[1], Calvin Wu[1], M. Charles Liberman[2], Susan E. Shore[1,3,4‡], Gabriel Corfas[1‡]*

1 Kresge Hearing Research Institute and Department of Otolaryngology—Head and Neck Surgery, University of Michigan, Ann Arbor, Michigan, United States of America, 2 Mass Eye and Ear Infirmary and Harvard Medical School. Boston, Massachusetts, United States of America, 3 Biomedical Engineering, University of Michigan, Ann Arbor, Michigan, United States of America, 4 Molecular and Integrative Physiology, University of Michigan, Ann Arbor, Michigan, United States of America

¤ Current address: Department of Otolaryngology, Peking University Shenzhen Hospital, Shenzhen, China
‡ These authors are joint senior authors on this work.
* corfas@med.umich.edu

**Data Availability Statement:** All data files are available from the Dryad database (accession number https://doi.org/10.5061/dryad.k6djh9w8v).

## Abstract

Loss of synapses between spiral ganglion neurons and inner hair cells (IHC synaptopathy) leads to an auditory neuropathy called hidden hearing loss (HHL) characterized by normal auditory thresholds but reduced amplitude of sound-evoked auditory potentials. It has been proposed that synaptopathy and HHL result in poor performance in challenging hearing tasks despite a normal audiogram. However, this has only been tested in animals after exposure to noise or ototoxic drugs, which can cause deficits beyond synaptopathy. Furthermore, the impact of supernumerary synapses on auditory processing has not been evaluated. Here, we studied mice in which IHC synapse counts were increased or decreased by altering neurotrophin 3 (Ntf3) expression in IHC supporting cells. As we previously showed, postnatal Ntf3 knockdown or overexpression reduces or increases, respectively, IHC synapse density and suprathreshold amplitude of sound-evoked auditory potentials without changing cochlear thresholds. We now show that IHC synapse density does not influence the magnitude of the acoustic startle reflex or its prepulse inhibition. In contrast, gap-prepulse inhibition, a behavioral test for auditory temporal processing, is reduced or enhanced according to Ntf3 expression levels. These results indicate that IHC synaptopathy causes temporal processing deficits predicted in HHL. Furthermore, the improvement in temporal acuity achieved by increasing Ntf3 expression and synapse density suggests a therapeutic strategy for improving hearing in noise for individuals with synaptopathy of various etiologies.

**Funding:** This work was supported in part by NIH/NIDCD R01 DC018500 (GC), NIH/NIDCD R01 DC000188 (MCL), and a grant from the American Tinnitus Association (GC). The funders had no role in study design, data collection and analysis, decision to publish, or preparation of the manuscript.

**Competing interests:** GC and MCL were scientific founders of Decibel Therapeutics, hadequity interest in the company and have received compensation for consulting. SES and DM are scientific founders of Auricle, Inc and have equity interest in the company. Neither company was involved in this study.

**Abbreviations:** ABR, auditory-brainstem response; AC, auditory cortex; ASR, acoustic startle response; BBN, broadband background noise; CNS, central nervous system; Ct, cycle threshold; DPOAE, distortion product otoacoustic emission; EDTA, ethylenediaminetetraacetic acid; GPIAS, gap-inhibition of the acoustic startle; HHL, hidden hearing loss; IC, inferior colliculus; IHC, inner hair cell; LGP, lateral globus pallidus; NBN, narrowband background noise; PnC, pontis caudalis; PPI, prepulse inhibition.

## Introduction

The inner hair cells (IHCs) of the mammalian organ of Corti transduce sounds into neural activity through glutamatergic excitatory synapses onto primary auditory neurons, the type I spiral ganglion neurons. These synapses are vulnerable to aging and noise exposure [1], i.e., even moderate noise exposure and/or normal aging result in loss of IHC synapses [2,3]. This type of deafferentation, a.k.a. cochlear synaptopathy, has been well documented in various mammalian species, including mice [1], guinea pigs [4], rats [5], gerbils [6], chinchillas [7], and humans [8,9]. Animal studies have shown that moderate cochlear synaptopathy caused by noise overexposure or aging causes "hidden hearing loss," i.e., an auditory neuropathy that manifests as a reduction in suprathreshold amplitudes of the first peak (peak I) of the auditory-brainstem response (ABR) without a shift in cochlear thresholds [1].

It has been hypothesized that cochlear synaptopathy leads to neural coding deficits that impair speech discrimination and intelligibility, especially in noisy environments, and to compensatory increases in the "gain" of central circuits that cause hyperacusis and tinnitus [10]. Indeed, experiments on animals with noise- or drug-induced cochlear synaptopathy show evidence for persistent gain in central circuits, either in the enhancement of acoustic startle responses [11], or of sound-evoked response rates in neurons from the inferior colliculus or cortex [12,13]. Cortical responses from synaptopathic animals also show threshold elevations for detectability of tones in noise without changes in tone detection in quiet [13]. However, other studies have failed to find such correlation [14]. In humans, several studies have shown a relationship between noise-exposure history and reductions in ABR suprathreshold amplitudes among those with normal audiometric thresholds [15], or a correlation between ABR peak I amplitudes and performance on speech-in-noise tasks [16,17], but these studies did not provide direct demonstration of cochlear synaptic loss. Furthermore, other studies have found no evidence for a relationship between noise history or ABR amplitudes and impaired speech perception [18].

To explore the impact of cochlear synapse density on auditory function, and to probe the effects of cochlear synaptopathy in the absence of cochlear insults like acoustic overexposure, ototoxic drugs or aging, where damage may be more widespread than the loss of cochlear synapses, we used transgenic mice in which IHC synapse density can be controlled via altering neurotrophin 3 (Ntf3) expression in the IHC's supporting cells [19]. We previously showed that altering the levels of supporting-cell Ntf3 expression starting in the neonatal period permanently changes IHC synapse density, i.e., supporting-cell Ntf3 overexpression increases IHC synapse numbers and enhances ABR peak I amplitudes, whereas supporting-cell-specific Ntf3 knock-out decreases them [19]. Here, we tested the behavioral phenotypes of these transgenic mice, focusing on the acoustic startle reflex and its modulation. For the latter, we measured prepulse inhibition (PPI) and gap-inhibition of the acoustic startle (GPIAS), behaviors used to assess stimulus salience, sensory gating, and temporal processing in both animals and humans [20–26].

We found that, compared to control animals, GPIAS is stronger in animals with increased IHC synapse density (and Ntf3 overexpression) and weaker in animals with decreased synapse density (and Ntf3 knockdown), without any changes in the strength of the acoustic startle itself or its PPI. As seen with noise- or age-related loss of synapses [5,11], the later ABR peaks, originating from auditory brainstem and midbrain nuclei, are unchanged after Ntf3 knockdown, despite a decrease in peak I amplitudes, suggesting a compensatory central gain. In Ntf3 overexpressors, ABR peaks I–IV are enhanced, suggesting there are no mechanisms for central gain reduction after a peripheral gain of function. Together, these results are consistent with the notion that enhancing IHC synapse density could lead to a general enhancement of temporal processing abilities.

## Results

### Regulating IHC synapse density and sound-evoked auditory nerve activity via Ntf3 knock-down or over-expression

To modify the levels of Ntf3 expression in IHC supporting cells, we used cell-specific inducible gene recombination as in prior studies [19]. Briefly, the Plp1-CreERT transgenic line [27] was used to drive gene recombination in IHC supporting cells via tamoxifen treatment during the neonatal period [28]. This CreERT transgene was combined with either conditional Ntf3 KO alleles (Ntf3$^{flox/flox}$) [29] to reduce Ntf3 expression in these cells, or an inducible Ntf3 overexpression transgene (Ntf3$^{STOP}$) to increase it [19]. Since Ntf3 is expressed by both IHCs and their surrounding supporting cells in the cochlea [30], knockout of the Ntf3 gene from IHC supporting cells in Ntf3$^{flox/flox}$::Plp1-CreER$^T$ reduces cochlear Ntf3 level but does not eliminate it [19]. As controls, we used mice with the conditional Ntf3 alleles without the CreERT transgene.

Between the ages of 8 and 15 weeks, mutant and control mice underwent a variety of behavioral and physiological tests (Fig 1). At 16 weeks, cochlear tissues were harvested to measure Ntf3 expression levels and the number of synapses per IHC (synapse density). We present the latter analyses first, which show that the Ntf3 manipulations had the expected effects on the cochlea.

Quantitative RT-PCR (Fig 2A and 2B) confirmed that cochlear Ntf3 levels were reduced in the Ntf3$^{flox/flox}$::Plp1-CreER$^T$, i.e., Ntf3 Knockdown (Ntf3-KD) mice and increased in the Ntf3$^{STOP}$::Plp1-CreER$^T$, i.e., Ntf3 Overexpressor (Ntf3-OE), mice. Similarly, expression of VGF mRNA, a gene downstream of neurotrophin receptor signaling [31], was decreased in the cochleas of Ntf3-KD mice (Fig 2A) and increased in Ntf3-OE mice (Fig 2B). Furthermore, there was a clear correlation between the mRNA levels of Ntf3 and VGF (Fig 2C), indicating that the changes in Ntf3 expression impact TrkC signaling in the inner ear. Since the Plp1 gene is also expressed in oligodendrocytes in the brain, and auditory-driven behaviors such as GPIAS are modulated by cortical circuits [32], we also measured the levels of Ntf3 and VGF in the cerebral cortex. We found a small decrease in cortical Ntf3 mRNA levels in Ntf3-KD mice (Fig 2D), no change in Ntf3-OE mice (Fig 2E), and most importantly, no changes in VGF mRNA levels in either of the mutants (Fig 2D and 2E), indicating that the manipulation of Ntf3 expression in Plp1-expressing cells is unlikely to have a direct effect on the central nervous system (CNS).

We also immunostained cochleas to assess the numbers of synapses between IHCs and auditory-nerve fibers. Whereas almost all auditory-nerve fibers contact a single IHC, each IHC is contacted by numerous auditory-nerve fibers. Each of these glutamatergic synaptic contacts can be identified as a closely apposed pair of puncta in cochleas immunostained for CtBP2, a major component of the presynaptic ribbon, and GluA2, a subunit of the glutamate receptors

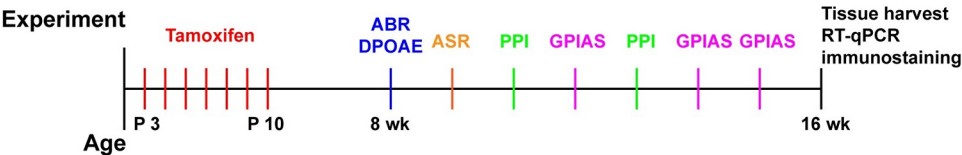

**Fig 1. Timeline of the experiments.** Experimental timeline showing the ages of mice for tamoxifen treatments, ABR measurements, behavioral assay (ASR, PPI, GPIAS), and sample collections for quantitative RT-PCR and immunostaining (P = postnatal day, wk = weeks). ABR, auditory-brainstem response; ASR, acoustic startle response; DPOAE, distortion product otoacoustic emission; GPIAS, gap-inhibition of the acoustic startle; PPI, prepulse inhibition.

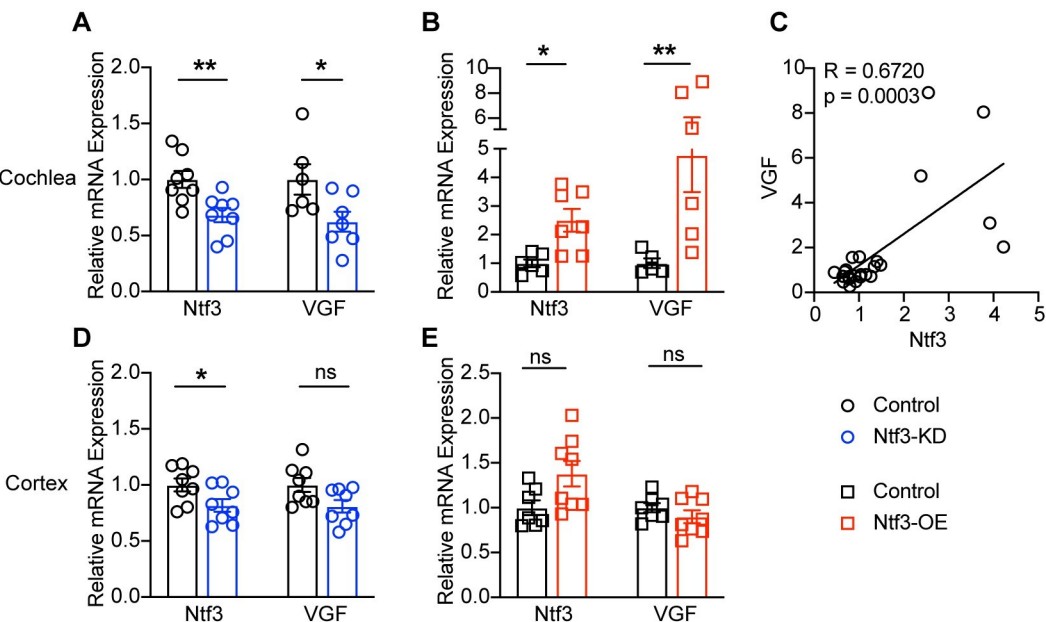

**Fig 2. Ntf3 expression in Plp1+ cells impacts TrkC signaling in the cochlea, not in the CNS.** mRNA level of Ntf3 and VGF, a gene downstream of TrkC signaling, are reduced in Ntf3-KD cochleas (**A**) and increased in Ntf3-OE cochleas (**B**). Furthermore, cochlear of Ntf3 and VGF mRNA levels are correlated (**C**). In contrast, cortical Ntf3 mRNA level is slightly decreased in Ntf3-KD mice (**D**) and unchanged in Ntf3-OE mice (**E**). No changes in VGF mRNA levels are observed in the brains of either Ntf3-KD or Ntf3-OE mice (**D, E**). $n = 6$–8, ns = $p > 0.05$, * $p < 0.05$, ** $p < 0.01$, mRNA levels were compared by two-tailed unpaired $t$ test. The data underlying this figure can be found in S1 Data. Error bars represent SEM. CNS, central nervous system.

localized at the postsynaptic terminals (Fig 3A and 3E). In a normal mouse, the mean number of synapses per IHC follows an inverted U-shaped function, peaking in mid-cochlear regions at a value of roughly 20 synapses per IHC [2,19]. As we previously reported [19], reduced supporting-cell Ntf3 expression levels decreased IHC synapse density in the basal half of the cochlea by as much as 20%, while increased supporting-cell Ntf3 expression levels increased IHC synapse density in the same cochlear regions by as much as 30% (Fig 3B–3D and 3F–3H). The stronger impact of supporting-cell derived Ntf3 on synapse density in the basal half of the cochlea likely reflects the fact that endogenous Ntf3 levels are lower in this region [30], making it more sensitive to changes in expression.

At 8 weeks of age, we assessed cochlear function by measuring distortion product otoacoustic emissions (DPOAEs), which reflect outer hair cell function, and ABRs, which reflect the summed responses of the auditory nerve and several higher auditory centers [33]. While peak I reflects synchronous sound-evoked activity of auditory-nerve fibers, the second peak (peak II) is dominated by contributions from cochlear-nucleus bushy cells, and later waves represent bushy-cell targets in the superior olivary complex and inferior colliculus [33–37]. These recordings confirmed that, as we reported earlier [19], reduced Ntf3 expression by IHC supporting cells reduces ABR peak I amplitudes without changing ABR and DPOAE thresholds (Fig 4A–4C), while increasing Ntf3 leads to normal thresholds with increased peak I amplitudes (Fig 4D–4F). An example raw recording of DPOAE and ABR waveform is shown in Fig 4G and 4H. Like the changes in synapse density, the effects of Ntf3 levels on peak I amplitudes are stronger in the middle and high frequencies, which reflect responses arising from the middle/basal cochlear regions.

Whereas peak I amplitude was reduced in mice with reduced Ntf3 cochlear expression, later ABR peak amplitudes were normal in these mice (Fig 5A), suggesting that decreased

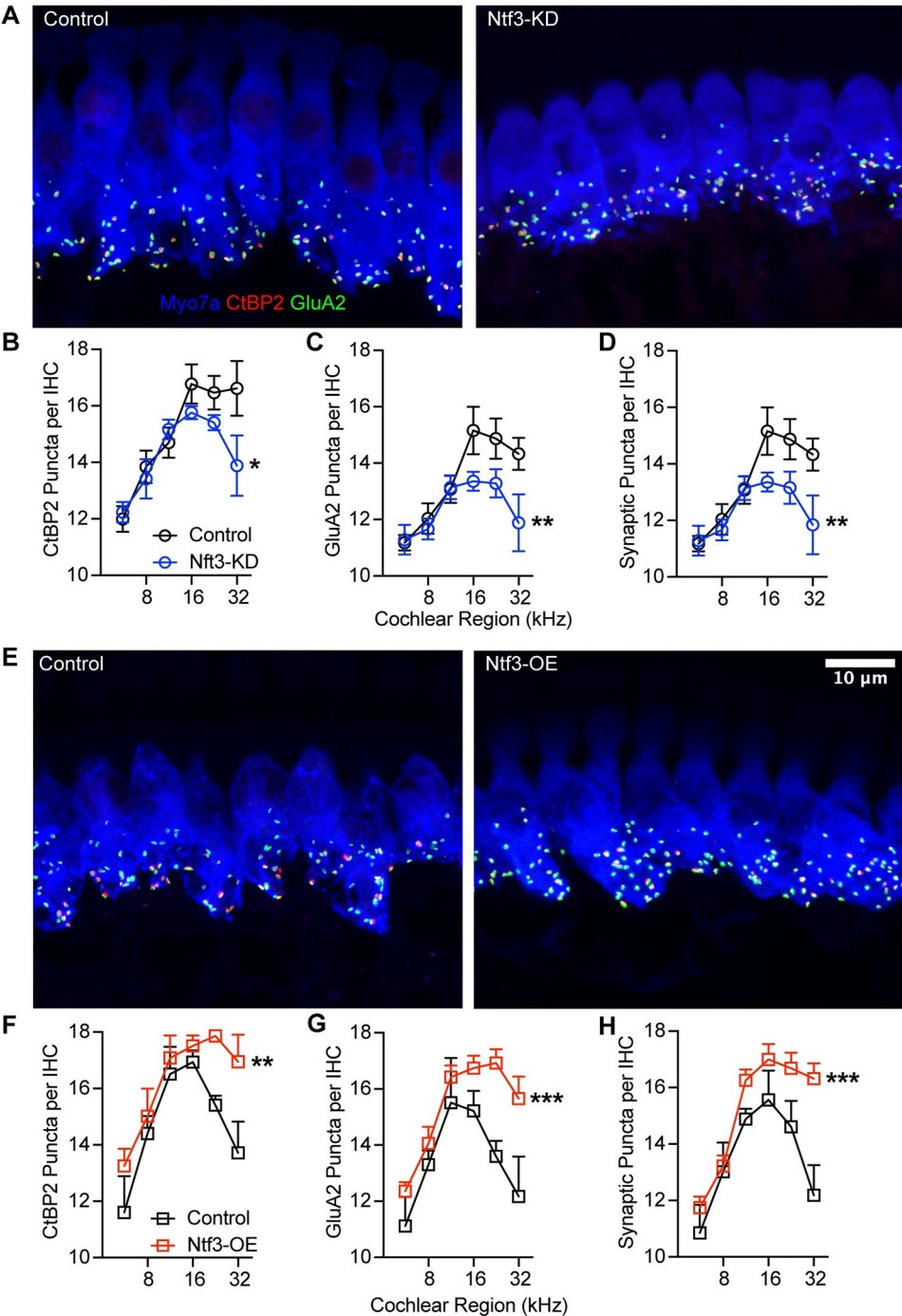

**Fig 3. Ntf3 regulates IHC synapse density.** Representative confocal images of IHC synapses at the 16 kHz cochlear region from Ntf3-KD (**A**) and Ntf3-OE (**E**) mice and their respective controls immunolabeled for presynaptic ribbons (CtBP2—red), postsynaptic receptor patches (GluA2—green), and hair cells (Myo7a - blue). Mean counts (± SEM) of ribbons (**B, F**), GluA2 patches (**C, G**), and colocalized markers (**D, H**) in Ntf3 KDs and OEs. $n$ = 5, ns = $p > 0.05$, * $p < 0.05$, ** $p < 0.01$, *** $p < 0.001$, *** $p < 0.0001$. Synaptic markers were compared by two-way ANOVA. The data underlying this figure can be found in S1 Data, the raw images were deposited in the Dryad repository (https://doi.org/10.5061/dryad.k6djh9w8v). Error bars represent SEM. IHC, inner hair cell.

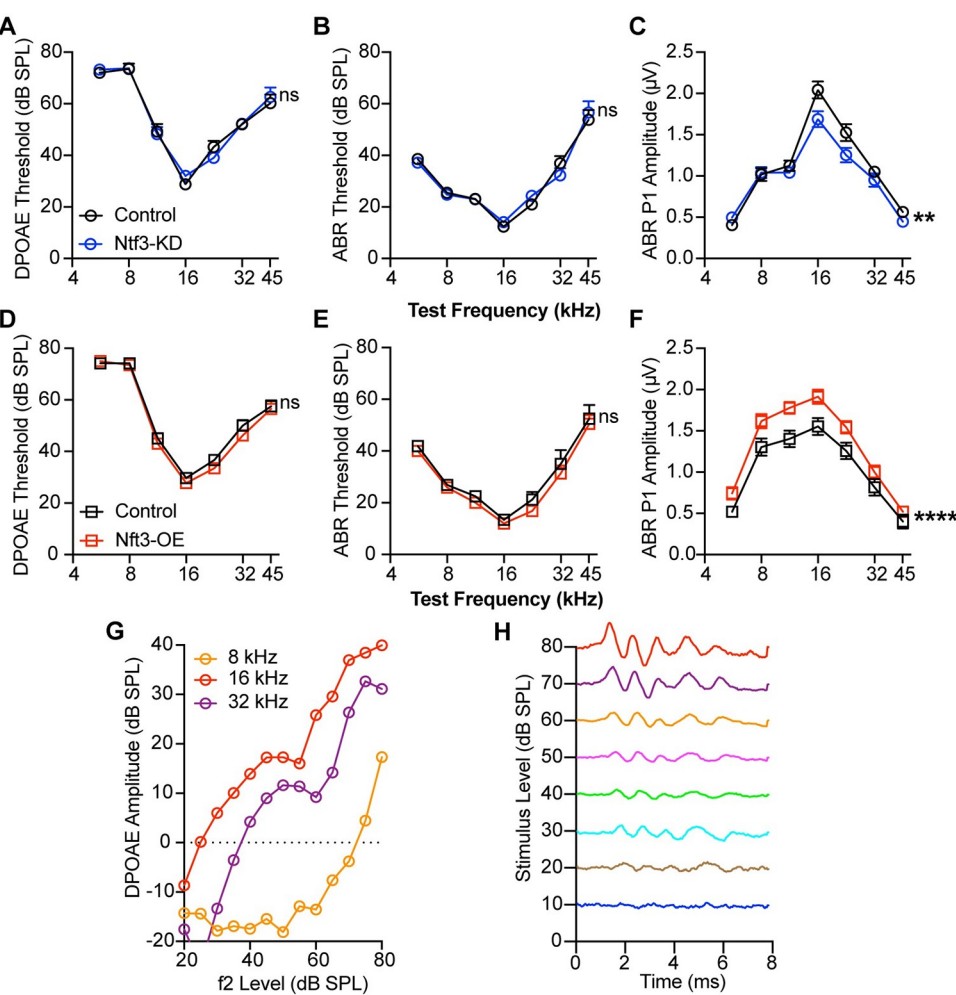

**Fig 4. Ntf3 knockdown or overexpression influence ABR peak I amplitudes without effecting cochlear thresholds.**
DPOAE (**A, D**) and ABR (**B, E**) thresholds in Ntf3-KD and Ntf3-OE mice are not different than their controls. In contrast, Ntf3 knockdown reduces ABR P1 amplitudes (**C**), whereas overexpression leads to increased peak I amplitudes (**F**). Representative traces of DPOAEs (**G**) and ABRs (**H**). $n$ = 15–24. ABR P1 amplitudes were assessed at 80 dB SPL. ns = $p > 0.05$, *$p < 0.05$, **$p < 0.01$, ***$p < 0.001$, ***$p < 0.0001$ by two-way ANOVA. The data underlying this figure can be found in S1 Data. Error bars represent SEM. ABR, auditory-brainstem response; DPOAE, distortion product otoacoustic emission.

sound-evoked activity of the auditory nerve leads to central gain in several higher auditory centers, as seen after noise-induced or age-related synaptic loss [5,11]. In contrast, Ntf3 over-expression resulted in increased amplitudes for ABR peaks I–IV (Fig 5B), indicating that increased IHC synapse density enhances sound-evoked signaling along the ascending auditory pathway. All peak latencies were normal in both mutants (Fig 6A and 6B), suggesting that auditory nerve myelination and conduction velocity was not affected by the altered Ntf3 levels and synapse densities [38].

## IHC synapse density influences auditory processing but not the startle reflex or sensory gating

To examine the impact of IHC synapse density on auditory processing, we tested 3 auditory-driven behaviors; the acoustic startle response (ASR) [39], and 2 behaviors that involve

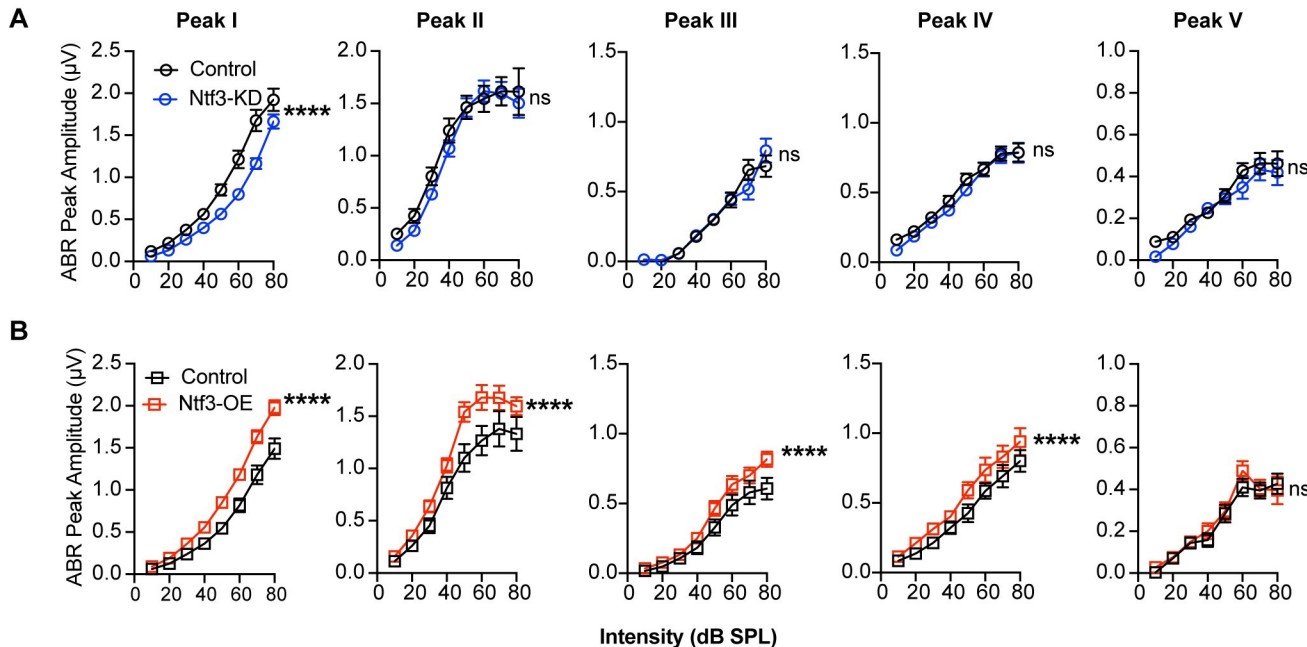

**Fig 5. Ntf3 knockdown or overexpression have different effects on the input-output function of ABR peaks I–IV.** Mean amplitude vs. level functions for ABR peaks I–IV in Ntf3-KD (**A**) and Ntf3-OE (**B**) mice and their respective controls at 16 kHz. Whereas peak I amplitudes are reduced in Ntf3-KD mice, the amplitudes of the other peaks remain normal, indicative of central compensation (**A**). In contrast, Ntf3 overexpression increases amplitudes of ABR peaks I to IV (**B**). $N = 14$–20, ns = p > 0.05, $*p < 0.05$, $**p < 0.01$, $***p < 0.001$, $***p < 0.0001$ by two-way ANOVA. The data underlying this figure can be found in S1 Data. Error bars represent SEM. ABR, auditory-brainstem response.

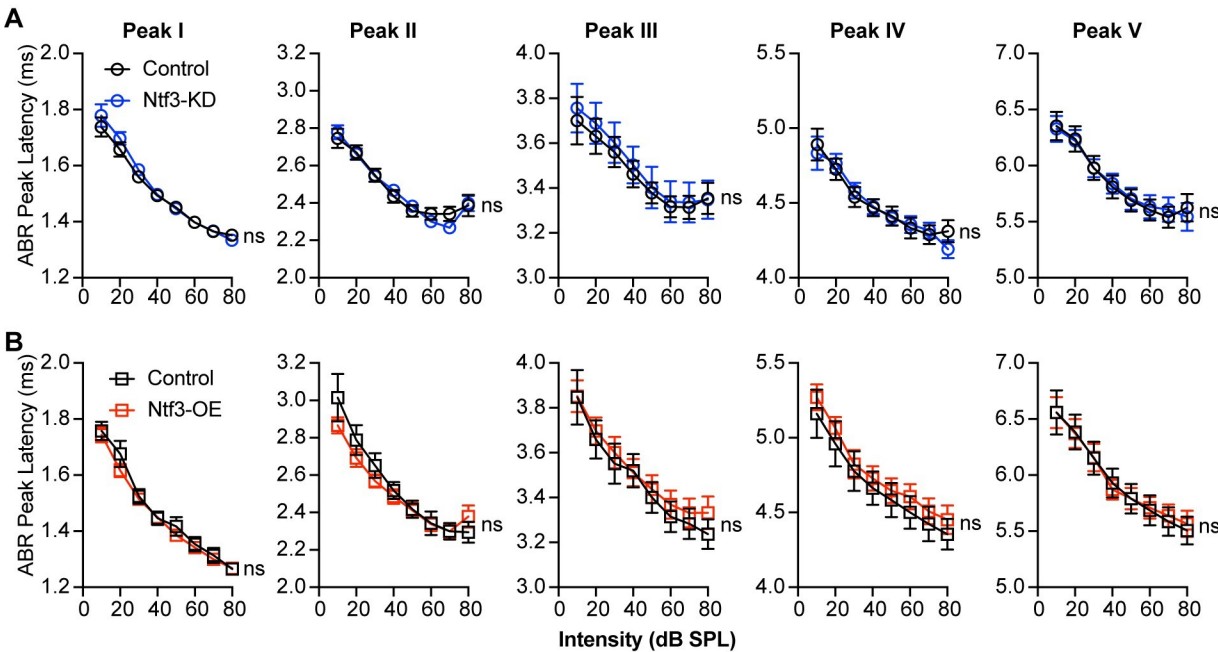

**Fig 6. Ntf3 expression levels do not influence the latencies of the ABR waveform peaks.** Plots of peak latency recorded at 16 kHz against sound stimulus level show that latencies of ABR peaks I–V are not altered by Ntf3-KD (**A**) and Ntf3-OE (**B**). $n = 14$–20, ns = $p > 0.05$ by two-way ANOVA. The data underlying this figure can be found in S1 Data. Error bars represent SEM. ABR, auditory-brainstem response.

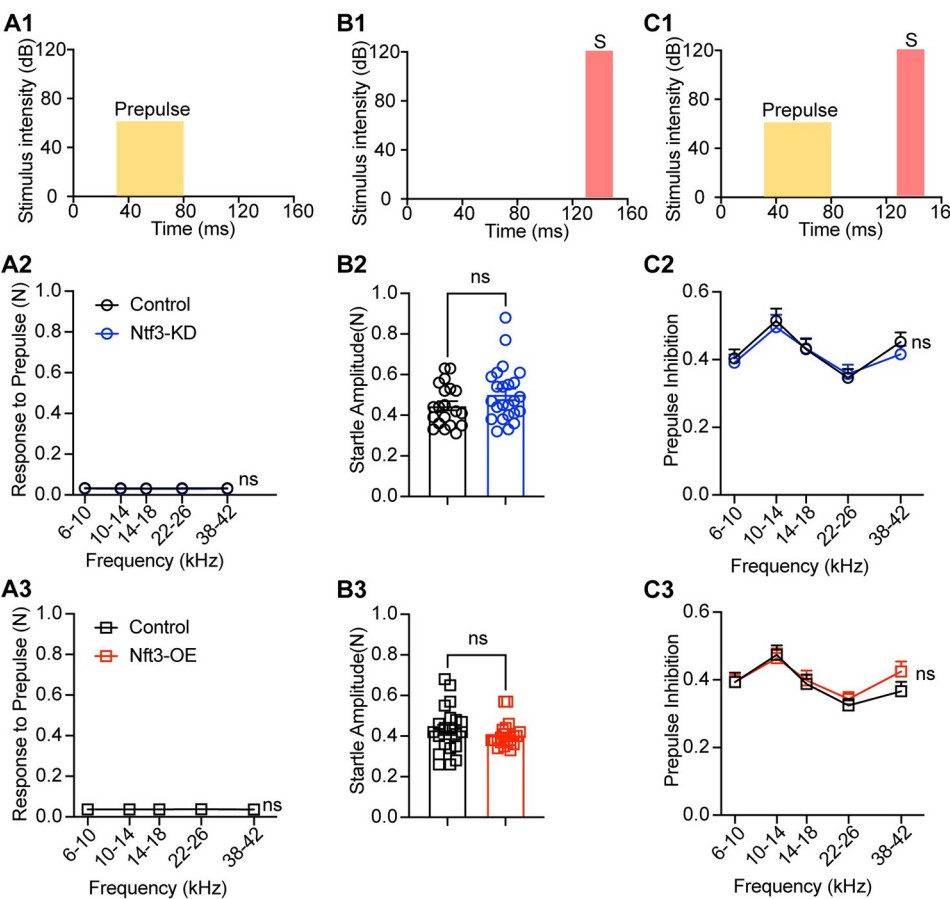

**Fig 7. IHC synapse density does not influence the acoustic startle response or prepulse inhibition.** Schematics of the protocols for prepulse stimulus (**A1**), startle stimulus (**B1**), and PPI stimuli (**C1**). (**A1**) The noise prepulse stimulus is a narrowband noise (4 kHz width around variable center frequencies, 65 dB SPL, 50 ms duration). (**B1**) The startle stimulus is a broadband noise (120 dB SPL, 20 ms duration). (**C1**) PPI consists of a noise prepulse and a startle stimulus that starts 50 ms after the prepulse. (**A2, A3**) Reactivity to prepulse is not significantly different between mutant and control mice. (**B2, B3**) Loud sound (120 dB SPL) elicits startle responses with amplitudes that were similar in mutant mice and their control littermates. (**C2, C3**) The degree of prepulse inhibition of the startle response by a prepulse was determined using the formula $PPI = 1 - \frac{startle\ response\ with prepulse}{startle\ response\ without\ prepulse}$. On average, the prepulse inhibit the startle response by 40%. There is no significant difference between control and mutant mice. $N$ = 20–24 mice/group for response to prepulse (**A2, A3**); $n$ = 18–20 mice/group for startle response (**B2, B3**); $n$ = 11–17 mice/group for PPI (**C2, C3**). ns = $p > 0.05$ by two-tailed unpaired $t$ tests (**B2, B3**) or two-way ANOVA (**C2, C3**). The data underlying this figure can be found in S1 Data. Mean ± SEM are shown. IHC, inner hair cell; PPI, prepulse inhibition.

modification of the ASR, prepulse inhibition of the ASR (PPI) [40], and gap-prepulse inhibition of the ASR (GPIAS) [41]. These tests have been a mainstay of studies on temporal processing and hearing-in-noise deficits in animal models [22,42–44].

The ASR is a reflexive and rapid burst of muscular activity in response to a sudden, brief, and intense sound. This is a robust and consistent behavior, easily quantified by measuring the whole-body startle response [45]. To investigate the impact of alterations in IHC synapse density on the ASR, we measured responses to moderate-intensity narrowband stimuli (50 ms 65 dB SPL) centered at different frequencies (8, 12, 16, 24, and 40 kHz) (Fig 7A1) and response to a high-intensity startle stimulus (20 ms 120 dB SPL broadband noise burst) (Fig 7B1), to detect hyper-responsiveness to innocuous sound and to determine if responses to the prepulse stimuli are altered by changes in Ntf3 levels. As expected, the moderate intensity prepulse

stimulus alone did not induce startle responses in either mutant or control mice (Fig 7A2 and 7A3). The high-intensity startle stimulus elicited strong responses with amplitudes that were not different between control and mutants (Fig 7B2 and 7B3). These results indicated that changes in Ntf3 expression, synapse density, and ABR peak I amplitudes do not affect reflexive motor responses to sound.

PPI is commonly used to assess sensorimotor gating, i.e., the ability of a sensory stimulus to suppress a motor response [26]. The PPI assay is quantified as the decrease in the ASR when a prepulse stimulus is presented a few milliseconds before the startle stimulus [46]. We used the 50 ms prepulse stimulus described above ending 50 ms prior to the startle stimulus (Fig 7C1) and quantified the magnitude of PPI as the fractional ASR reduction, i.e., 1 minus the ratio of startle magnitude with and without prepulse [46]. On average, the prepulse inhibited the startle response by 40%. There was no significant difference between control and mutant mice in PPI (Fig 7C2 and 7C3), indicating that changes in synapse density and associated changes in auditory-nerve activity do not affect sensorimotor gating.

Finally, we measured GPIAS, which is used to examine auditory temporal processing [20,22,25,47] and correlates with speech recognition in humans [48,49]. In GPIAS, the suppression of the ASR is induced by the presentation of a brief silent gap in a continuous background noise instead of a mild sound stimulus in silence. Gap detection was measured by presenting animals in broadband background noise (BBN) with gaps of various durations (3 to 50 ms) ending 50 ms before the startle stimulus (Fig 8A). Importantly, the startle amplitudes in background noise were normal in mice with altered Ntf3 expression (Fig 8B and 8C), indicating that synapse number does not alter the salience of the startle stimulus in the presence of background noise.

As done by others [14,22], we quantified GPIAS as the fractional reduction of startle, i.e., 1 minus the ratio of the startle magnitude with and without a gap ($Gap\ inhibition = 1 - \frac{startle\ response\ with\ gap}{startle\ response\ without\ gap}$). Analysis of gap inhibition as a function of gap duration (Fig 8D and 8E) showed that, consistent with previous reports [22], GPIAS is stronger with longer gaps. More importantly, Ntf3-KD mice showed a significant decrease in the gap inhibition, whereas Ntf3-OE mice showed a significant increase compared to their respective controls, indicating that IHC synapse density influences this modification of the ASR. Importantly, gap inhibition for both genotypes was stable across different sessions (S1 Fig), indicating that the results were not influenced by the age of the mice withing the 8 weeks of testing. Furthermore, there was a strong correlation between gap inhibition and ABR P1 amplitudes (S2 Fig), providing evidence that the magnitude of the sound-evoked auditory potentials is critical for the GPIAS.

The conclusion that IHC synapse density influences GPIAS was also supported by 2 additional methods of quantitative analysis of the GPIAS, gap detection threshold and Rd'. Gap detection threshold, which is used to measure the temporal acuity for acoustic transients, is defined as the gap duration that elicits 50% of the maximal inhibition level [22,42,50–54]. As for gap inhibition, gap-detection threshold was higher in Ntf3-KD mice (18.65 ± 3.473 ms) compared to their controls (8.38 ± 1.413 ms), and lower in Ntf3-OE mice (6.712 ± 0.8304 ms) compared to their controls (10.92 ± 1.775 ms) (Fig 8F and 8G). Similar conclusions could be reached by analysis of Rd', ($Rd = \frac{startle\ response\ without\ gap - response\ with\ gap}{standard\ deviation\ of\ gap\ conditions}$), a parameter that reflects the salience of each gap condition for each mouse [55]. As shown in panels 8H and 8I, analysis of Rd' as a function of gap duration showed similar trends as for gap inhibition, i.e., Ntf3-KD mice showed a significant decrease in the Rd' curve, whereas Ntf3-OE mice showed a significant increase compared to their respective controls (Fig 8H and 8I).

Since the spectral components and bandwidth of background noise also affects the behavioral gap detection [55], we presented gaps of 50-ms duration when mice were subjected to

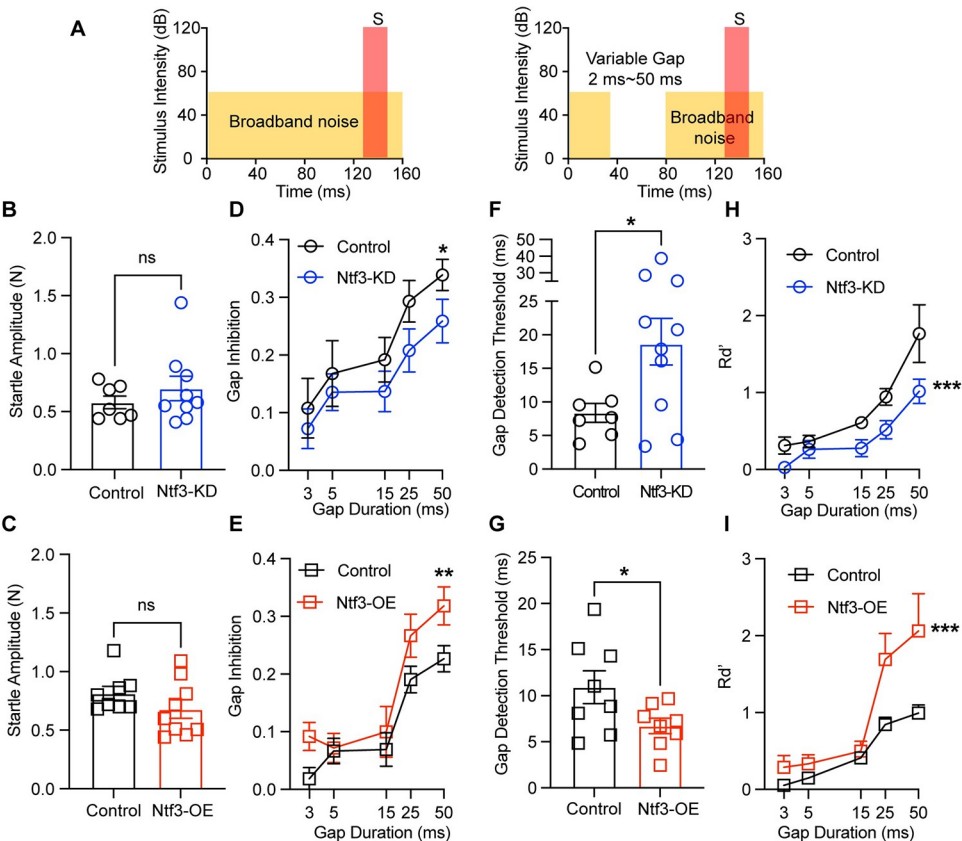

**Fig 8. Ntf3 expression levels influence gap detection thresholds in broadband background noise.** (**A**) Schematic depiction of NO-GAP trials (left) and GAP trials (right). NO GAP trials consisted of a startle sound (120 dB SPL, 20 ms duration) presented in continuous noise background (broadband noise, BBN, 65 dB SPL). In contrast, in the GAP trials, a silent gap in the background noise of variable length (0–50 ms) was presented ending 50 ms before the startle stimulus (S). (**B, C**) ASR amplitudes for the NO-GAP trials were similar in Ntf3 mutant mice and their control littermates. (**D, E**) Show the level of gap inhibition vs. gap length and for Ntf3 KD and OE mice, respectively. The inhibition of the startle reflex increases as the gap duration increases. (**F, G**) Show gap detection thresholds. Gap detection threshold is increased in Ntf3-KD mice (**H**) and decreased in Ntf3-OE mice (**I**) compared to their littermate controls. (**H, I**) Show level of Rd' vs. gap length for Ntf3 KD and OE mice, respectively. $n = 7$–20 mice/group, *$p < 0.05$, **$p < 0.01$, ***$p < 0.001$, ***$p < 0.0001$ by two-tailed unpaired $t$ test (**B, C, H,** and **I**) or two-way ANOVA (**D, E, F,** and **G**). The data underlying this figure can be found in S1 Data. Mean ± SEM are shown. ASR, acoustic startle response; BBN, broadband background noise.

narrowband background noise (NBN) centered around different frequencies (6 to 10 kHz, 10 to 14 kHz, 14 to 18 kHz, 22 to 26 kHz, and 38 to 42 kHz; spectral width: 4 kHz) (Fig 9A). Whereas Ntf3 expression levels did not influence the startle responses at any frequency (Fig 9B and 9C), the mutant mice showed consistent differences with the littermate controls with NBN at all frequencies (Fig 9D and 9E). These results indicate that the difference in suppression level caused by altered Ntf3 expression cannot be attributed to impact on startle amplitudes, and that the influence of Ntf3 expression levels on gap inhibition level is not restricted to any specific frequency.

## Discussion

Self-reporting studies indicate that 10% to 20% of adult humans with normal audiometric thresholds have hearing difficulties [56–59]. Furthermore, humans with sensorineural hearing loss, particularly in the high frequencies, have longer gap-detection thresholds that correlate

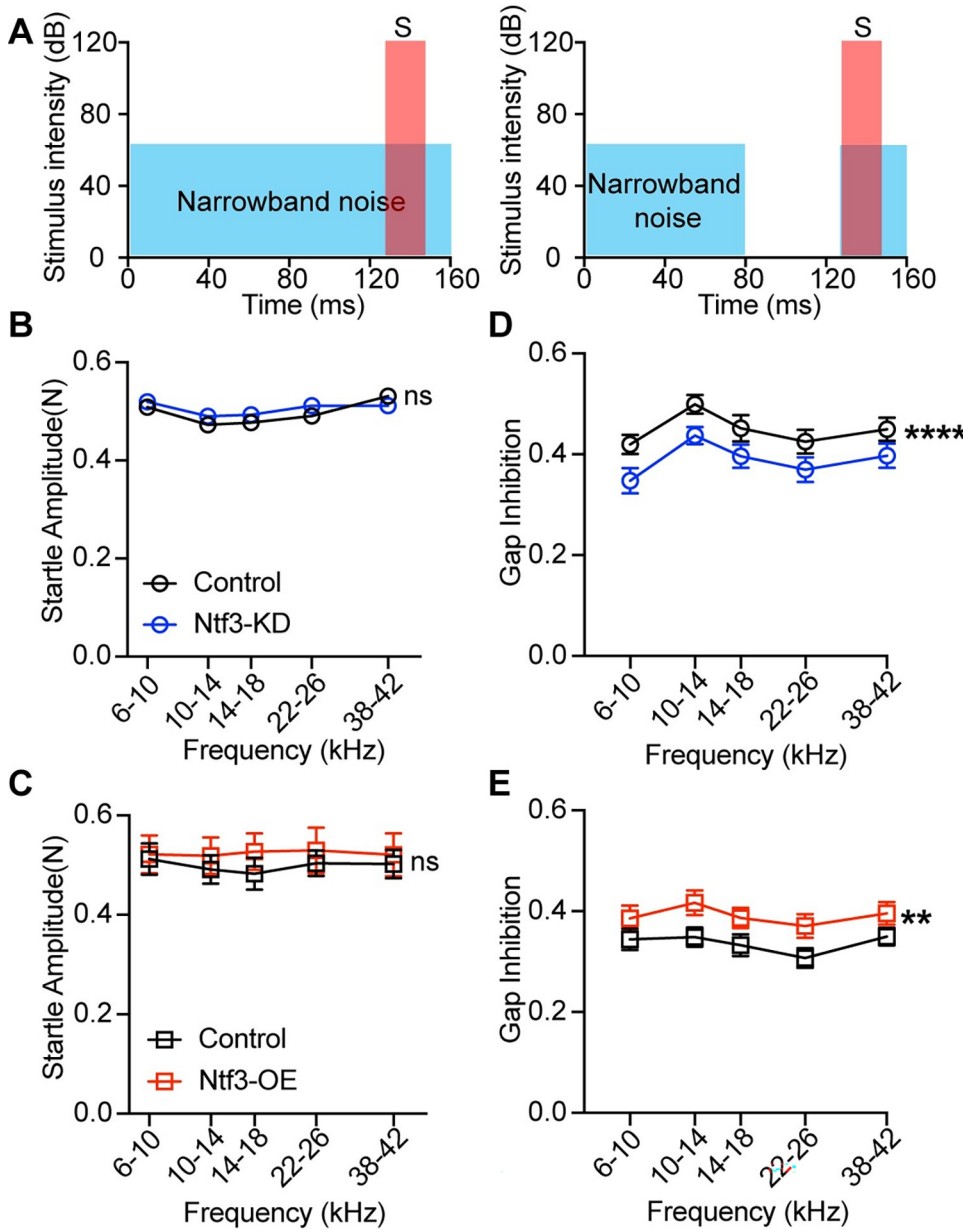

**Fig 9. Ntf3 expression levels influence gap inhibition in NBN.** (**A**) A schematic view of No gap trials (left) and gap trials (right). No gap trials consist of a startle sound (120 dB SPL, 20 ms duration) presented in continuous noise background (narrowband noise, NBN, 4 kHz width around variable center frequencies, 65 dB SPL). Gap-prepulse inhibition of the acoustic startle (GPIAS) was tested in gap trials with the same background noise with a 50-ms gap included as a prepulse followed 1 ms later by the startle-eliciting stimulus. (**B, C**) Responses to startle stimulus in continuous NBN background are unaffected in the Ntf3-KD or Ntf3-OE mice. (**D, E**) GPIAS in narrowband noises were significantly weakened in Ntf3-KD mice and strengthened in Ntf3-OE mice in a two-way ANOVA. Sidak multiple comparison tests revealed a significant reduction at frequency of background noise band 10–14, 14–18, and 22–26 kHz in Ntf3-KD mice and no frequency-specific changes in Ntf3-OE mice. $n$ = 10–14 mice/group, *$p < 0.05$, **$p < 0.01$ by two-way ANOVA. The data underlying this figure can be found in S1 Data. Mean ± SEM are shown. GPIAS, gap-inhibition of the acoustic startle; NBN, narrowband background noise.

with poorer speech perception [60–64]. It has been proposed that cochlear synaptopathy could contribute to this highly prevalent auditory dysfunction [3,11,15]. However, since synaptopathy in living human subjects cannot be assessed directly, and the impact of synapse density on

auditory processing had not been specifically tested in animal models without confounders such as noise-exposure or aging, the causal link between IHC synaptopathy and hearing impairments was previously hypothetical. Our findings using mice with reduced or enhanced IHC synapse density due to changes in Ntf3 expression provide direct evidence that auditory processing is influenced by cochlear synapse density and that IHC synaptopathy is sufficient to degrade auditory processing and temporal acuity. Furthermore, our results suggest that increased IHC density leads to enhanced temporal acuity, raising the possibility that therapies that increase IHC synapse density could improve auditory performance in humans.

Analysis of ABR waveforms in our mouse models provides new insights into the impact of IHC synapse density on the function of the ascending auditory pathway. We found that ABR peaks II–V are normal in mice with IHC synaptopathy induced by Ntf3 knockdown despite the reduced peak I amplitude. This finding is similar to that in animals with cochlear damage [3,11–13,65], which has been interpreted as reflecting homeostatic compensation in the auditory brainstem and midbrain after partial peripheral deafferentation. Our results provide further support for this view. Most surprising was the observation that ABR peak I–IV amplitudes are increased in mice with supernumerary IHC synapses. This finding indicates that increased IHC synapse density enhances sound-evoked signaling along the ascending auditory pathway. Our prior work showed that Ntf3-induced increase in synaptic counts is not associated with a proliferation of auditory nerve fibers [19]. Thus, we presume that a fraction of fibers now must make more than just 1 synaptic contact with an IHC. This, in turn, might lead to better synchronization of onset responses and higher ABR amplitudes, which are then propagated through the ascending central circuitry. It is possible that increased neural synchrony contributes to the improved detection of noise gaps in the Ntf3 overexpressing mice, suggesting that creating supernumerary IHC-SGN synapses could also improve auditory performance in a wider range of stimulus contexts.

Since the Plp1-CreERT transgene also drives gene recombination in oligodendrocytes in the CNS [27], and auditory cortical circuits contribute to gap detection [32], it was important to determine if the mice had significant changes in Ntf3 expression and signaling in the CNS. Finding little or no change in Ntf3 and VGF expression in the cortex of Ntf3 KD and OE mice, together with data showing that few CNS neurons express the Ntf3 receptor TrkC in the brain (see Expression Summary in [66]) suggests that the Ntf3 manipulations we created did not directly affect cortical circuits. Thus, the phenotypes seen here are most likely dominated by changes originating with the altered IHC synapse density.

Interestingly, mice with IHC synaptopathy due to Ntf3 knockdown did not show enhanced magnitude either of the baseline acoustic startle or PPI. This contrasts with observations in mice with synaptopathy induced by noise [11] or aging [46], where the hyper-responsiveness has been interpreted as a sign of hyperacusis, a clinical disorder associated with noise-induced or age-related hearing loss, in which moderate-intensity sounds become intolerably loud [23,46,67–69]. There are several potential explanations for this apparent discrepancy. First, whereas the noise- or age-induced acoustic trauma occurs after the circuits are fully formed, the synapse reduction in our model originates during the neonatal period, when central circuits are still developing, raising the possibility that some aspects of auditory processing adapt to the early onset of the synaptopathy, preventing the development of hyper-responsiveness. Second, whereas the extent of synaptic loss in our model is 20% to 30%, the other studies involved more extensive cochlear damage [11,46], which could lead to more dramatic central dysfunction. Third, noise exposure induces stress or fear responses, which can play a role in hyperacusis-like behavior [70].

Alterations in IHC synapse density influence performance in GPIAS but not in PPI, likely reflecting different neural mechanism underlying these sound-driven modifications of the

ASR, which are mediated by higher order nuclei [71,72]. During ASR, the auditory-nerve inputs activate cochlear root neurons that project to nucleus reticularis pontis caudalis (PnC) neurons, which then excite the spinal motor neurons that elicit the whole-body startle response [73,74]. Prepulses and gaps are relayed from the cochlear root neurons to the inferior colliculus (IC) [75], but their pathways diverge thereafter. Prepulse processing has been shown to involve the lateral globus pallidus (LGP), whereas gap processing involves the auditory cortex (AC) [54,76]. Thus, our results suggest that alterations in synapse density might affect primarily the latter. Further studies on the impact of synapse density on these circuits could provide new insights into the mechanisms by which peripheral responses influence central auditory processing.

Published gap-detection thresholds for mice vary among studies, e.g., some reported them to be around 2 ms [42,77,78], other 4 ms [32], or 8 ms [22], the latter being similar to our findings. Several factors might contribute to this discrepancy, including the age of the animals [42], the spectral components and bandwidth of the background noise [55], the gap and startle onset/offset ramp [53], and the restrainer setting [79].

We previously demonstrated that increasing cochlear Ntf3 availability after noise exposure induces IHC synapse regeneration and recovery of ABR peak I amplitudes [19,80,81]. More recently, we showed that increasing cochlear Ntf3 expression levels in the middle-aged mouse acutely increases ABR peak I amplitudes and slows the progression of age-related synaptopathy, preserving the IHC synapse density of middle age until the end of the lifespan [82]. The results from the current study suggest that Ntf3-based therapies could not only promote IHC synapse health and numbers, but also improve auditory processing after noise trauma or in aging.

In summary, our finding that cochlear synaptopathy elicited without cochlear insults such as noise or ototoxic drugs results in temporal processing deficits further supports the notion that synaptopathy is a key contributor to the impaired speech perception experienced by many with hidden hearing loss. Furthermore, the improvement in temporal acuity achieved by increasing Ntf3 expression and synapse density suggests a therapeutic strategy for improving hearing in noise for individuals with synaptopathy of various etiologies.

## Materials and methods

### Animals

All experimental procedures complied with the National Institutes of Health guidelines and were approved by the Institutional Animal Care and Use Committee of University of Michigan, Michigan, United States of America (PRO00011287). Cochlear supporting-cell-specific Ntf3 knock-down mice (Ntf3-KD, tamoxifen-treated $Ntf3^{flox/flox}$::$Plp1/CreER^T$ mice) and Ntf3 overexpressing mice (Ntf3-OE, tamoxifen-treated $Ntf3^{stop}$::$Plp1/CreER^T$ mice) were generated as previously described [19]. $Ntf3$-$KD$ mice and their controls ($Ntf3^{flox/flox}$) were maintained on C57BL/6 background that carry the wild-type allele of Cdh23, and therefore do not have accelerated age-related hearing loss. $Ntf3$-$OE$ and their controls ($Ntf3^{stop}$) were on FVB/N background. Both male and female mice were included in this study (Dryad DOI: https://doi.org/10.5061/dryad.k6djh9w8v).

### Tamoxifen administration

Tamoxifen was injected into intraperitoneal cavity of P 3–10 $Ntf3^{stop}$::$Plp1/CreER^T$ mice or P 1–3 $Ntf3^{flox/flox}$::$Plp1/CreER^T$ mice as previously described [19]. A 10 mg/ml solution of tamoxifen was obtained by dissolution in corn oil. Injection was 33 mg/kg for $Ntf3^{stop}$::$Plp1/CreER^T$ mice and 50 mg/kg for $Ntf3^{flox/flox}$::$Plp1/CreER^T$ mice.

## Real-time quantitative RT-PCR

Total RNA was isolated from the cortical brain and cochlea samples from 1-month-old mice using RNA extraction kit and QIAzol Reagent (RNeasy mini kit; Qiagen, Germany), and DNase treatment was performed (RNase-free; Qiagen). The complementary DNA was synthesized using iScript cDNA synthesis kit (Bio-Rad, #1708891, USA), according to the manufacturers' protocol. Quantitative RT-PCR was performed on a CFX-96 Bio-Rad reverse transcription polymerase chain reaction detection system (Hercules, California, USA) using iTaq Universal SYBR Green supermix (Bio-Rad, # 172–5121, USA) and primer pairs were synthesized by IDT (Coralville, Iowa, USA). All samples and standard curves were run in triplicate. Water instead of complementary DNA was used as a negative control. The 10 μl reaction contained 5 μl of SYBR Green supermix, 6 pmol of each forward and reverse primer (0.6 μl), 1.9 μl nuclease-free of water, and 2.5 μl of cDNA sample. The mRNA expression levels in Ntf3$^{flox/flox}$::Plp1/CreER$^T$ or Ntf3$^{STOP}$::Plp1/CreER$^T$ versus their Ntf3$^{flox/flox}$ or Ntf3$^{STOP}$ control counterparts were determined by a comparative cycle threshold (Ct) method and relative gene copy number was calculated as normalized gene expression, defined as described previously [83]. Ribosomal protein L19 (RPL19) was used as the housekeeping gene. The following specific oligo primers were used for the target genes: Rpl19, F: 5′ACCTGGATGAGAAGGA TGAG 3′; R: 5′ACCTTCAGGTACAGGCTGTG 3′; Ntf3, F 5′GCCCCCTCCCTTATAC CTAATG 3′; R: 5′CATAGCGTTTCCTCCGTGGT 3′; Vgf: F: 5′GGTAGCTGAGGACGC AGTGT 3′; R: 5′GTCCAGTGCCTGCAACAGT 3′. Changes in mRNA expression were calculated as relative expression (arbitrary units) respective to the control group for each mouse line.

## Immunostaining and synaptic counts

Cochleas from 16-week-old mice were prepared for whole-mount imaging as described in [19]. In brief, the samples were fixed with 4% formaldehyde for 2 h and then decalcified with 5% ethylenediaminetetraacetic acid (EDTA) for 3 to 5 days. Cochlear epithelia were micro-dissected into 5 segments for whole mount processing. The cochlea segments were permeabilized by freeze–thawing in 30% sucrose and then blocked with 5% normal horse serum for 1 h. Afterwards, the following primary antibodies were used: (1) CtBP2 to visualize synaptic ribbons (mouse anti-CtBP2 at 1:200, BD Biosciences, catalog # 612044, RRID: AB_399431); (2) GluR2 to visualize postsynaptic receptors (mouse anti-GluR2 at 1:2,000, Millipore, catalog # MAB397, RRID: AB_2113875); and (3) Myosin VIIa to visualize IHCs (rabbit anti-Myosin VIIa at 1:200; Proteus Biosciences, catalog # 25–6790, RRID: AB_10015251). Secondary antibodies used were Alexa Fluor 488 conjugated anti-mouse IgG2a (1:1,000, Invitrogen), Alexa Fluor 568 conjugated anti-mouse IgG1 (1:1,000, Invitrogen), and Alexa Fluor 647 conjugated anti-rabbit (1:1,000, Life Technologies). Frequency maps were created using a custom ImageJ plug-in. Images were captured from 5.6 to 45.2 kHz using a Leica SP8 with a 1.4 NA 63× oil immersion objective at 3 digital zoom. Offline image analysis was performed using Amira (Visage Imaging). Quantification of synapses was done by an investigator blinded to experimental groups.

## Distortion product otoacoustic emissions (DPOAEs) and auditory brainstem responses (ABRs)

DPOAEs and ABRs were performed as previously described [19]. Mice were anaesthetized by i.p. injections of xylazine (20 mg kg$^{-1}$, i.p.) and ketamine (100 mg kg$^{-1}$, i.p.). The DPOAEs were elicited by 2 primary tones (f1 and f2) and recorded at $(2 \times f1)-f2$. f1 level was 10 dB

higher than the f2 level and frequency ratio f2/f1 was 1.2. The ear-canal sound pressure was amplified and averaged at 4 μs intervals. DPOAE thresholds were defined as the f2 level that produced a response 10 dB SPL higher than the noise floor. For ABR measurement, subdermal electrodes were placed (1) at the dorsal midline of the head; (2) behind the left earlobe; and (3) at the base of the tail (for a ground electrode). ABRs were evoked with 5 ms tone pips (0.5 ms rise–fall) delivered to the eardrum. The frequencies of tone pips were 5.6, 8, 11.3, 16, 22.6, 32, and 45.2 kHz, with 15 sound levels from 10 to 80 dB SPL for each frequency. The signals were amplified 10,000 times and filtered through a 0.3 to 3 kHz passband. At each level, the average of 1,024 signals was taken after "artifact rejection." Both recordings were performed using National Instruments input/output boards hardware. Offline analysis was performed using Excel and custom ABR peak-analysis software that finds inflection points in the waveform, subject to correction by an experienced observer.

For microphone and probe-tube calibration during ABR and DPOAE measurement, a brief sound containing frequencies throughout the range to be calibrated is produced by one of the earphones, voltage out of the acoustic-assembly microphone is measured while the sound pressure near the probe-tube is measured simultaneously with a reference microphone. From the results and the characteristics of the reference microphone, the system computes the ratio of the voltage out of the acoustic-assembly microphone to the SPL at the end of the probe-tube.

ABR threshold was determined by visual analysis of stacked waveforms from highest to lowest SPL. Threshold was determined as the lowest level at which a repeatable Wave I could be identified. Wave I–V amplitude was defined as the difference between a 1-ms average of the pre-stimulus baseline and the wave I–V peak, after additional high-pass filtering to remove low-frequency baseline shifts. For DPOAE, pressure measurements in the ear canal were averaged with spectral and waveform averaging, then the amplitudes of the DPOAE responses at 2f1-f2 then were analyzed as input-output functions. All thresholds were determined by 2 observers, including one blinded and experienced.

## Pre-pulse inhibition (PPI) and gap inhibition of the acoustic startle (GPIAS)

Mice were tested in a $10 \times 4.5 \times 4$ cm cage inside a sound-isolation chamber that was placed within a sound attenuating room. The sound source was located in the upper part of this chamber. The piezoelectric motion sensor attached to the cage detected the vertical force of the startle reflex. All ASR, PPI, and GPIAS stimuli and responses were generated and recorded with Kinder Scientific Startle Monitor (Kinder Scientific, Poway, California, USA).

PPI tests were used for assessing sensorimotor gating on 8- to 15-old-week mice, twice a week, each 2 days apart. PPI tests were conducted quiet. The startle stimuli were BBN bursts at 120 dB SPL, 20 ms in duration, 0.1 ms rise-fall times. The prepulse was a narrow-band sound centered at 8, 12, 16, 24, and 40 kHz, 50 ms in duration, with 2-ms rise-fall ramps (Fig 7). PPI test consisted of prepulse trials and startle-only trials, which were delivered alternatively. In prepulse trials, a prepulse ended 50 ms before the startle stimulus. Startle-only trials were similar to the prepulse trials, but no prepulse was delivered. PPI startle ratio is the ratio of the startle magnitude in prepulse trials over the startle magnitude in startle-only trials.

The GPIAS paradigm has been used for measuring auditory temporal processing [20,25]. Gap inhibition was assessed on 8- to 15-old-week mice, twice a week, 2 days apart. The testing consists of 2 types of trials, gap trials and no-gap trials (Figs 8A and 9A) that were delivered alternatively. In both trials, the startle stimulus was 20 ms BBN at 120 dB with 0.1 ms rise/fall times. The startle was preceded either by gaps with varied durations (3-, 5-, 15-, 25-, or 50-ms long) embedded in BBN or by a 50-ms gap embedded in a narrow-band background sound

centered at 8, 12, 16, 24, and 40 kHz at 65 dB. Sensor calibration is accomplished with a 100 g calibration weight. For sound-level calibration, the external ¼" microphone (B&K type 4136) is mounted on top of the sensing plate and connected to a spectrum analyzed (Stanford Research Systems, Model SR760).

For each mouse, PPI and gap-startle ratios were averaged from 11 sessions. Each session of PPI and gap-PPI test included 60 pairs of prepulse and startle-only trials for PPI or gap and no-gap trials for gap detection (5 prepulse or background sound frequencies, 12 pairs for each frequency). The interval between trials was randomly varied between 5 and 15 s. Each session began with a 2-min acclimatization period in the cage before startle testing began, and all tests were conducted in darkness. Startle-only trials and no-gap trial amplitudes greater than or equal to mean ± 2.5 standard deviations were eliminated. When a trial was eliminated, its paired trial was also eliminated. About 8% to 13% of the trials were eliminated based on these criteria. PPI ratios were calculated as the average with prepulse startle amplitude divided by the mean without prepulse startle amplitude. We analyzed the GPIAS using gap inhibition ($inhibition = 1 - \frac{startle\ response\ with\ gap}{startle\ response\ without\ gap}$), Rd' (Rd = $\frac{startle\ response\ without\ gap - response\ with\ gap}{standard\ deviation\ of\ gap\ conditions}$), and/or gap threshold method. For analysis of gap threshold, the data were fitted with a three-parameter logistic function: $f(x) = \frac{d}{1 + \exp(b(\log(x) - \log(e)))}$. Recordings with a fit coefficient ($R2$) below 0.6 were excluded from the analysis [22]. The gap-detection threshold was considered as the value of the fitted curve that elicited 50% of the maximal inhibition.

## Statistical analyses

Analyses were performed using GraphPad Prism 6 (GraphPad Software Inc., La Jolla, California, USA) and RStudio packages. Data are shown as mean and standard error of the mean (SEM). The number of replicates (*n*) is indicated in the results section and figure legends. No explicit power analysis was used to predetermine sample sizes, but our sample sizes are similar to those reported in our previous publications. Statistical differences in auditory physiology (DPOAE threshold, ABR threshold, amplitude, and latency), ribbon synapse counts, behavioral background movement, PPI ratio, gap-startle ratios were analyzed using two-way ANOVA, followed by Bonferroni multiple comparisons test. mRNA expression, ASR amplitude, and gap detection threshold were compared using unpaired Student's *t* test. Correlations were computed using Pearson's correlation. Statistical threshold was set to alpha = 0.05.

Dryad DOI

https://doi.org/10.5061/dryad.k6djh9w8v [84]

## Supporting information

**S1 Fig. Gap inhibition is stable across different sessions for both genotypes.** The relationship between gap inhibition vs. gap length does not change between the 3 time points for Ntf3-KD and Ntf3-OE mice. Mean ± SEM are shown. Summary data displayed in S1A–S1D Fig can be found in S1 Data.
(TIF)

**S2 Fig. The degree of gap inhibition correlates with ABR peak I.** The amplitude of ABR peak I versus gap inhibitory level of Ntf3-KD and their littermate controls (A) or Nft3-OE and their littermate controls (B) show a linear correlation. Summary data displayed in S2A and S2B Fig can be found in S1 Data.
(TIF)

**S1 Data. Raw data used for the generation of the graphs presented in Figs 2–9 and S1–S2.**
(XLSX)

## Author Contributions

**Conceptualization:** Lingchao Ji, David T. Martel, M. Charles Liberman, Susan E. Shore, Gabriel Corfas.

**Data curation:** Lingchao Ji, Beatriz C. Borges.

**Formal analysis:** Lingchao Ji, David T. Martel, Calvin Wu, Gabriel Corfas.

**Funding acquisition:** Susan E. Shore, Gabriel Corfas.

**Investigation:** Lingchao Ji, Beatriz C. Borges.

**Methodology:** Lingchao Ji, David T. Martel, Calvin Wu, Susan E. Shore, Gabriel Corfas.

**Supervision:** Susan E. Shore, Gabriel Corfas.

**Writing – original draft:** Lingchao Ji, Gabriel Corfas.

**Writing – review & editing:** Lingchao Ji, Beatriz C. Borges, David T. Martel, Calvin Wu, M. Charles Liberman, Susan E. Shore, Gabriel Corfas.

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
