## [Editor Report · Decision Letter 0]

5 May 2023

Dear Dr Corfas, 

Thank you for submitting your manuscript entitled "Inner hair cell synapse density influences auditory processing" for consideration as a Research Article by PLOS Biology.

Your manuscript has now been evaluated by the PLOS Biology editorial staff as well as by an academic editor with relevant expertise and I am writing to let you know that we would like to send your submission out for external peer review.

Once your full submission is complete, your paper will undergo a series of checks in preparation for peer review. After your manuscript has passed the checks it will be sent out for review. To provide the metadata for your submission, please Login to Editorial Manager (https://www.editorialmanager.com/pbiology) within two working days, i.e. by May 09 2023 11:59PM.

Kind regards,

Ines

--

Ines Alvarez-Garcia, PhD

Senior Editor

PLOS Biology

---

## [Decision Letter · Decision Letter 1]

12 Jul 2023

Dear Dr Corfas,

Thank you for your patience while your manuscript entitled "Inner hair cell synapse density influences auditory processing" was peer-reviewed at PLOS Biology. Please also accept my apologies for the delay in providing you with our decision. The manuscript has been evaluated by the PLOS Biology editors, an Academic Editor with relevant expertise, and by three independent reviewers. 

As you will see, the reviewers find the conclusions of the manuscript interesting and significant for the field, but they also raise several points that would need to be addressed to strengthen the results. The reviewers think that additional analyses of startle responses in broadband noise should be performed to support the main conclusions and that the variability in gap detention thresholds should be also addressed, among other issues.

In light of the reviews and consultation with the Academic Editor, we would like to invite you to revise the work to thoroughly address the reviewers' reports. Given the extent of revision needed, we cannot make a decision about publication until we have seen the revised manuscript and your response to the reviewers' comments. Your revised manuscript is likely to be sent for further evaluation by all or a subset of the reviewers.

**IMPORTANT - SUBMITTING YOUR REVISION**

3. Resubmission Checklist

a) *PLOS Data Policy*

b) *Published Peer Review*

Sincerely,

Ines

--

Ines Alvarez-Garcia, PhD

Senior Editor

PLOS Biology

Reviewers' comments

Rev. 1:

The manuscript presents interesting results indicating that increasing the number of cochlear ribbon synapses improves detection of silent gaps in noise, whereas reducing the number of synapses I commend the authors for incorporating behavioral assays into their work, since tone-evoked ABRs may not always be good predictors of more complex aspects of sound processing. However, there are several issues that should be addressed to improve confidence in the main findings.

Previous psychoacoustic experiments indicate that good high frequency hearing supports sensitivity to short gaps in noise. Gap detection thresholds are increased (worsened) with high frequency sensorineural hearing loss or with lowpass noise when measured behaviorally using traditional psychoacoustic methods. Though this previous work is not thoroughly addressed in the manuscript, it follows that an increased number of ribbon synapses in the higher frequency cochlear regions could improve processing of gaps as reported in this study. It is not surprising that PPI and GPIAS startle inhibition patterns were different given that typical tone/noise-burst PPI utilizes different circuits than GPIAS, as the authors acknowledge. However, there is some additional data analysis of startle responses in broadband noise that should be performed to support the central conclusion. Additionally, the variability in gap detection thresholds--even in the control groups—should be addressed, as typical gap detection thresholds for mice are on the order of ~2ms, not the >10ms reported here for about 2/3 of the mice. Clarification of these issues is essential to verifying the central claim that "IHC synapse density influences auditory processing but not the startle reflex or sensory gating." There are also some methodological clarifications that would strengthen the manuscript.

Methods:

The calibration procedures for the various hearing tests are not described.

Ribbon synapse counts were determined by a blind observer. ABR and DPOAE thresholds should similarly be determined by a blind observer. Additionally, the authors should describe the threshold determination method.

Please provide additional information about how the ABR analysis software identified peaks. Was this analysis supervised by an experienced observer?

How many startle trials +/- 2SD and large PPI or gap startle ratios were eliminated (proportion)?

Results:

We cannot assume that ABR and DPOAE thresholds remained normal in control or KD/OE groups over the 8 week time period between ABR/DPOAE testing and the final gap detection tests. Hearing status, particularly high frequency hearing, may have been affected over this time with repeated exposure to loud startle-eliciting stimuli and background noise during testing.

Since startle-eliciting stimulus salience is affected by the presence of background noise, the authors should show the ASR amplitudes for the no-gap (broadband noise only) conditions for all groups to confirm that baseline ASR is not different under these conditions. If there is a group difference, then the relationship between ASR in noise (no gap) and gap-induced inhibition should be examined, because there is evidence in mice that prepulse inhibition is related to the strength of the ASR. Increased ASR in noise indicates that the startle-eliciting stimulus is more salient, and this could affect the amount of inhibition that is possible (for example, Csomor et al. 2008). A related issue is that startle-eliciting stimuli presented at saturation for an individual mouse does not produce optimal inhibition (see Longenecker and Galazyuk). A 120 dB startle-eliciting stimulus is almost certainly at the point of response saturation.

What is the rationale behind choosing 50% ASR inhibition as the gap detection threshold? This is not the 'conventional' method as stated in the manuscript. Previous research has employed several different methods to calculate gap detection thresholds (such as Rd', minimum gap that produces statistically significant inhibition, etc). More sensitive thresholds might be obtained using a different threshold estimation procedure.

Both control groups showed extremely variable gap detection thresholds. About half of the Ntf3-KD mice showed gap detection thresholds that were similar to controls, and gap detection thresholds varied immensely in this group from ~2-50 ms. The reasons for this performance variability should be addressed. For the Ntf3-OE comparison, the control groups' gap detection thresholds (Fig 8E) varied more than the Ntf3-OE mice. Why were some animals able to perform well while others were not? Gap detection thresholds for 'normal-hearing' mice (and most other species tested) are typically ~2ms when measured using GPIAS, traditional operant psychoacoustic methods, and physiological recordings in the inferior colliculus and auditory cortex (e.g. Walton et al.,1997, Ison et al. 2005; Radziwon et al., 2009; Weible et al. 2014). A comparison of the present results to previous gap detection measurements in mice should at least be addressed in the Discussion. One possible explanation for the discrepancy is that the acoustics of the experimental setup were such that the noise offset/offset received by the mouse was not sharp due to echos or restrainer issues, and this may have varied depending on where the individual mouse's head was located (see Ison et al. 2002; Longenecker and Galazyuk 2012).

Additionally, gap detection thresholds measured using startle modification procedures can improve over time due to learning (e.g. Fitch et al. 2008). The authors should compare the GPIAS results over the three test timepoints to determine if there are differences in learning that cause improvements (or potentially habituation) to the gaps.

Minor Issues:

Line 426: freeze-thawing in 30% of what solution? There seems to be a missing word.

Line 489: Unclear why reference # 73 is cited here. No GPIAS or PPI experiments were reported in that paper.

Rev. 2: Neil J. Ingham – note that this reviewer has signed his review

The authors present a well written paper on the effects of altering the numbers of inner hair cell – auditory nerve fibre synapses on auditory physiological responses and suprathreshold behavioural measurements of auditory processing in the mouse. The explosion of interest in cochlear synaptopathy over a decade or more as a potential mechanism to explain hearing problems despite a normal audiogram has, to some extent, been hampered by the means of inducing synaptopathy in animal models. The need to used old-aged animals, or animals exposed to noise or ototoxins, to investigate synaptopathy has always left residual doubts over the exact explanation of the results and data caused by secondary consequences of the induction process. Lots of data exist that correlate any particular effect on auditory function and processing with reduction in cochlear synapse numbers, but these correlations may well be tainted by other pathological processes occurring at the same time. The model developed by the authors over recent years to use over- and underexpression of neurotrophin-3 as a means to up- and downregulate synapse numbers under the inner hair cells is potentially a much “cleaner” and more selective method to investigate synaptopathy in isolation from other secondary effects of inducing synaptopathy by more aggressive means.

The work presented here builds on previous work from this group in characterisation of the Ntf3 system on synaptopathy and provides important results that even in young animals, auditory processing as a result of cochlear synapse changes can be altered in a way consistent with the direction of synapse number change. The results present a potential basis for the development of tools to help diagnose cochlear synaptopathy in humans, which will be of interest to those working in clinical settings and auditory psychophysicists and pharmaceutical / gene therapy development.

The main findings that synapse density does not affect acoustic startle or it’s prepulse inhibition, but that density does influence the gap-prepulse inhibition test of auditory processing is a very interesting and novel result, to the best of my knowledge. The finding that a peripheral synaptopathy, apparently without central synaptic effects, can have a profound effect of the higher functions of auditory processing and behavioural outcomes is quite remarkable.

The authors provide a thorough introduction and background to this study, pointing out gaps in previous knowledge that they are attempting to fill. The methods used are robust, well established and performed appropriately for the context. Statistical comparisons seem to have been performed with care and appropriately for the data. The data and results are presented clearly and unambiguously. The gene expression results in figure 2 show up and down -regulation of expression in the knockdown and over-expression mouse models used. The synaptic distribution and densities described in figure 3 are nicely consistent with the 2 mouse models. As always, the confocal images of the hair cells and synaptic components are a pleasure to look at! The measurements of auditory sensitivity in figure 4 clearly show “normal hearing” in the 2 mutants. The ABR wave amplitude and latency plots in Figure 5&6 are nicely presented. The data in figure 7 clearly support the authors description of no changes in the magnitude of the acoustic startle or its prepulse inhibition. The gap-detection thresholds presented in Figure 9 are clearly modulated in either direction depending of the direction of synapse density modification, which is also apparent in the gap inhibition vs noise band plots in figure 10.

The authors present a clear and concise discussion of their results, providing appropriate explanations of the findings in the context of the synaptopathy field. The discussion on the likely mechanisms by which the peripheral synaptopathy (and sensory nerve input) affects a behavioural output is welcome and highlights anatomical pathways in the brain that are not perhaps as widely considered by auditory researchers as they should be.

Some comments on specific points:

Ln 121-133. The description of figure 2 and labelling of panels appears to be different in the main text and in the figure legend. In the text, the VGF-Ntf3 correlation is referred to as Fig 2C, but in the legend and on the figure itself, this plot is Fig 2E. The authors should check the figure layout and correct the main text and/or legend accordingly.

Ln 144-158 & Figure 3. The pattern of synaptic component counts and density vs cochear region in control mice appears somewhat different – especially in the more basal regions – between the 2 mouse lines. The control curves (in black) are more symmetrical in the Nft3-OE control mice and less symmetrical in the Nft3-KD control mice. Would the authors like to comment on these differences? Is it perhaps related to the different gene

---

## [Decision Letter · Decision Letter 2]

12 Mar 2024

Dear Dr Corfas,

Thank you for your patience while we considered your revised manuscript entitled "Inner hair cell synapse density influences auditory processing" for publication as a Research Article at PLOS Biology. This revised version of your manuscript has been evaluated by the PLOS Biology editors, the Academic Editor and one of the original reviewers.

The review is attached below. Based on this review and our Academic Editor's assessment of your revision, we are likely to accept this manuscript for publication, provided you satisfactorily address the policy-related requests stated below.

In addition, we would like you to consider a suggestion to improve the title:

"An increase of cochlear inner hair cell synapses improves temporal auditory processing in mice"

We expect to receive your revised manuscript within two weeks. 

*Published Peer Review History*

*Press*

Sincerely,

Ines

--

Ines Alvarez-Garcia, PhD

Senior Editor

PLOS Biology

ETHICS STATEMENT:

Thank you for providing the ethics statement. Please also include an approval number.

Fig. 2A-E; Fig. 3B-D, F-H; Fig. 4A-H; Fig. 5A, B; Fig. 6A, B; Fig. 7A1-C3; Fig. 8A-I; Fig. 9A-E; Fig. S1A-D and Fig. S2A, B

***Please also make publicly available at this stage the data deposited in the Dryad database (accession number https://doi.org/10.5061/dryad.k6djh9w8v).

Reviewers' comments

Rev. 1: Amanda Lauer - note that this reviewer has signed her review

Thanks for providing a more thorough consideration of your interesting behavioral results.

---

## [Editor Report · Decision Letter 3]

7 May 2024

Dear Dr Corfas,

Thank you for the submission of your revised Research Article entitled "From hidden hearing loss to supranormal auditory processing by neurotrophin 3-mediated modulation of inner hair cell synapse density" for publication in PLOS Biology. On behalf of my colleagues and the Academic Editor, Jennifer Bizley, I am delighted to let you know that we can in principle accept your manuscript for publication, provided you address any remaining formatting and reporting issues. These will be detailed in an email you should receive within 2-3 business days from our colleagues in the journal operations team; no action is required from you until then. Please note that we will not be able to formally accept your manuscript and schedule it for publication until you have completed any requested changes.

PRESS

Sincerely, 

Ines

--

Ines Alvarez-Garcia, PhD

Senior Editor

PLOS Biology
